# Factors Affecting Mechanical Properties of Reinforced Bioplastics: A Review

**DOI:** 10.3390/polym14183737

**Published:** 2022-09-07

**Authors:** Jet Yin Boey, Chee Keong Lee, Guan Seng Tay

**Affiliations:** 1Bioresource Technology Division, School of Industrial Technology, Universiti Sains Malaysia, Penang 11800, Malaysia; 2Bioprocess Technology Division, School of Industrial Technology, Universiti Sains Malaysia, Penang 11800, Malaysia; 3Green Biopolymer, Coatings & Packaging Cluster, School of Industrial Technology, Universiti Sains Malaysia, Penang 11800, Malaysia

**Keywords:** mechanical properties, physical treatment, chemical treatment, biological treatment

## Abstract

The short life cycle and recalcitrant nature of petroleum-based plastics have been associated with plastic waste accumulation due to their composition rather than worldwide overproduction. The drive to replace single-use products has sparked a considerable amount of research work to discover sustainable options for petroleum-based plastics. Bioplastics open up a new horizon in plastics manufacturing operations and industrial sectors because of their low environmental impact, superior biodegradability, and contribution to sustainable goals. Their mechanical properties regarding tensile, flexural, hardness, and impact strength vary substantially. Various attempts have been made to augment their mechanical characteristics and capacities by incorporating reinforcement materials, such as inorganic and lignocellulosic fibres. This review summarizes the research on the properties of bioplastics modified by fibre reinforcement, with a focus on mechanical performance. The mechanical properties of reinforced bioplastics are significantly driven by parameters such as filler type, filler percentage, and aspect ratio. Fibre treatment aims to promote fibre–matrix adhesion by changing their physical, chemical, thermal, and mechanical properties. A general overview of how different filler treatments affect the mechanical properties of the composite is also presented. Lastly, the application of natural fibre-reinforced bioplastics in the automobile, construction, and packaging industries is discussed.

## 1. Introduction of Bioplastics

Plastics represent a broad category of polymer composites that constitute polymers as a building block. Polymeric materials can be divided into either thermoplastic (which softens when heated and stiffens again when cooled) or thermosetting polymers (which do not soften when they have been moulded). Most thermoplastic and thermosetting materials in present industrial use are petroleum-derived and non-renewable, posing a limitation to the polymer industry [1,2]. As a result, plastic waste has been primarily portrayed as a plastic composition issue rather than a global overproduction problem. Thankfully, plastics can be more sustainable through the convergence of technology improvements and consumer preferences, making them more achievable than ever. Bioplastic is the foundation of the principle of sustainable development, from exploiting more renewable content and recycling materials, to lowering manufacturing energy and returning material to nature at the end of its life. They reduce the pollutants induced by petroleum-derived plastics that remain solid for centuries, signalling a new era of packing technology and industry. The demand for bioplastics has gone through the roof in both industry and research. This is because people are worried about pollution in the environment more than ever, and local and international groups are passing stricter laws to protect the environment.

Bioplastics are a type of plastic material that is bio-based, biodegradable, or both, depending on the source from which they were created. To put it another way, the term bioplastic also refers to petroleum-based plastics that are biodegradable. It can be any combination of bio-based (partially, completely, or non-biobased), biodegradable, or compostable, provided that it is not both non-bio-based and non-biodegradable. “Bio-based” is termed as products created from biological material derived from biomass such as plants, bacteria, algae, etc. [1,3]. For instance, in conventional plastics (i.e., petroleum-derived and non-biodegradable), the traditional petrochemical resin is replaced by biopolymers extracted from animals or plants, while synthetic glass or carbon fibres are substituted by natural fibres like jute, bamboo, flax, and hemp [4,5,6,7]. The word “biodegradability” refers to a broad range of enzymatic and/or chemical reactions mediated by bacteria or biological organisms, the efficiencies of which are governed by the conditions in which these polymers biodegrade [8]. Microorganisms, industrial or home composting as an end-of-life option, as well as anaerobic digestion, may also decompose bioplastics, encouraging a more sustainable circular economy [9]. The substitution of petroleum-based feedstock with renewable feedstock provides an extra benefit as it relies less on fossil fuel as the carbon source. Nevertheless, it does not imply that the need for fossil fuels is eliminated. With this, the amount of greenhouse gas emissions associated with bioplastic production is reduced. Because the carbon dioxide (CO_2_) taken from the air during photosynthesis compensates for the CO_2_ released during biodegradation, it can be carbon neutral or even carbon negative [10,11].

With emerging innovation, it is now possible to design more sustainable plastics with distinct physical and aesthetic properties to compete with conventional plastics like polystyrene (PS), polypropylene (PP), and polyethylene terephthalate (PET). New bio-based materials have the increasingly popular ability to minimise environmental concerns while addressing the existing polymer and composite demand [1]. Bioplastic can be tailored to behave similarly to traditional plastics in the manufacturing phase but also excel from a performance standpoint. Currently, bioplastics account for only a relatively low proportion of global plastics production [10]. The cost of bioplastics is the main concern for the future extent of implementation as commercial manufacturing processes are expensive [8]. In this context, manufacturing costs can be cut down by integrating organic waste and residues, thus lowering the number of biodegradable polymers required to make bioplastics [2]. Metabolic and genetic engineering advancements have led to microbial and plant strains that may considerably boost yields and production capacities while being cost-effective [12]. When these considerations are paired with the conservation of scarce fossil fuels and the increment of environmental consciousness, it is foreseeable that the market dominance for bioplastics may develop in the future and replace petroleum-based single-use products, such as containers, straws, cups, and cutlery.

## 2. Types of Bioplastics and Process of Moulding Different Types of Bioplastics

Bioplastics, unlike conventional plastics, are mostly derived from renewable raw materials, including vegetable fats, oils, whey, starch, cellulose, and chitosan [1,3,10]. They are classified into two groups based on their backbone chemical composition: bio-based plastics and biodegradable plastics.

Bio-based plastics can be either biodegradable or non-biodegradable. Aliphatic polyesters like poly(lactic acid) (PLA), polyhydroxyalkanoates (PHA), starch, and cellulose are examples of both bio-based and biodegradable plastics. PLA is a thermoplastic biodegradable polyester that is produced through the polymerisation of bio-derived monomers, such as corn, potato, sugarcane, etc. It is recognised as one of the most popular “green” polymers in the polymer market, extensively used in food packaging applications and the biomedical sector [13]. In addition, PHA is an aliphatic bioplastic synthesised naturally by bacteria through the fermentation of lipids and sugar [14]. Besides utilising natural resources such as glucose, starch, and edible oils as the substrate for PHA production, several academics have investigated the potential of employing industrial, agricultural, and food waste, along with wastewater [12,15,16,17]. Starch is a biodegradable polysaccharide polymer that is widely used in food packaging applications owing to its abundance, low material cost, and food safety [18]. To be treated as a deformable thermoplastic polymer, a plasticiser (urea, glycerol, or sorbitol) with the addition of water to produce thermoplastic starch (TPS) under elevated temperature. TPS can then be extruded to make foam and solid moulded objects [19]. Cellulose, a polysaccharide composed of β-D-glucose subunits, is another biodegradable polymer [11].

As previously mentioned, the fact that they are bio-based does not imply that they are inherently biodegradable; that is, they contain renewable or fossil-fuel-based carbon [1]. For example, bio-polyethylene terephthalate (bio-PET), polyethylene-2,5-furandicarboxylate (PEF), and bio-polyethylene (bio-PE) are chemically identical to fossil-based PET and polyethylene (PE) [20]. This plastic type accounts for more than 42% of global bioplastic production capacities [3]. Polyamides (PA) represent another example of non-biodegradable bioplastics with high mechanical strength used in medical implants [21]. Over the years, attention has turned to PEF, a new polymer expected to hit the global market by 2023. PEF is like PET, but it is made entirely of bio-based materials and has better barrier properties, making it a great choice for bottles of drinks [10].

Aside from that, there are hydrocarbon thermoplastics that can be produced from renewable resources to replace a portion of the monomer, one of which is poly(1-butene) (PB). Cui et al. [22] synthesised isotactic poly(1-butene) (iPB) from eugenol, which is a phenol compound that can be extracted from different types of plant oil, such as clove oil, laurel oil, and camphor oil. The monomer of 1-butene and eugenol were copolymerized in the presence of Ziegler-Natta as a catalyst. The results indicated that the introduction of eugenol in the synthesis of poly(1-butene) has improved the thermal stability of the product and prevented the thermo-oxidative reaction of the polymer chain.

To specify whether the material is biodegradable or compostable, specific standards and protocols are required, which are standardised by the International Organization for Standardization (ISO), the American Society for Testing and Materials (ASTM International), government institutions, and other associations [1,23]. Their biodegradability is primarily determined by their physical and chemical structures but also by the environmental conditions in which they are placed [10,23]. Biodegradable polymers are typically derived from biological sources, but they can also be derived from petroleum resources. Biodegradable but non-biologically derived polymers include poly(butylene adipate-co-terephthalate) (PBAT), polycaprolactone (PCL), polyvinyl alcohol (PVA), and polybutylene succinate (PBS) [13].

There are different approaches for producing green composites, including injection moulding (IM), compression moulding (CM), extrusion, calendaring, thermoforming, and resin transfer moulding (RTM) [24,25,26]. Broadly, the selection of the processing techniques is a trade-off between the processing time, production cost, and final product design, shape, and size. The technique chosen affects the fibre dispersion, orientation, and aspect ratio, hence defining the mechanical properties [7,27]. Additionally, the processing conditions, such as temperature, pressure, and speed, vary from one technique to another. Factors, such as fibre length, content, type, and moisture content, can also affect the manufacturing process. For instance, there is a possibility that fibre would be thermally degraded if the temperature used is too high; therefore, it is preferable for those matrices that have melting points lower than the degradation temperature [26]. In view of this, it is essential to use the right methods and parameters to get the best results when making composites.

Injection moulding is a closed moulding process, which involves injecting a material into a mould under elevated temperatures and pressure. This technique is suitable for fabricating metal, glass, thermoplastic or thermosetting materials into assorted sizes and shapes of plastic products within a short time with high precision [21,28,29]. This process necessitates the use of an IM machine (equipped with a hopper, plunger, a heating unit, and a clamping system), raw plastic material, and a mould or die. The cycle starts with the injection stage, followed by the holding and plasticising stages, and finally the ejection of the moulded component. During the injection stage, the injection moulding compound (IMC) is introduced into the injection chamber via the feed hopper. After that, IMC will be conveyed by a screw-type plunger into a heated barrel, which transforms it into liquid form. It is then mechanically transferred into the closed mould cavity to cool and acquire the desired shape. During the holding and plasticising stages, the mould is kept at a constant temperature and pressure so that the melt can solidify rapidly after it is filled. Once the compound is hardened, the mould plates will open and eject the finished part via ejector pins. A new cycle can be started at this stage [28,29,30]. Regrettably, the significant expense of producing the mould is frequently a hurdle to IM technology. This method works best for the large-scale mass production of identical products [31].

Compression moulding is a traditional manufacturing technique that involves pressing thermoset and thermoplastic materials in the form of granules, sheets, or prepregs between two matched metal dies with huge presses [30]. The short cycle time and high production rate of CM make it ideal for applications in the automobile industry [7,27]. The common intermediate materials are sheet moulding compound (SMC), bulk moulding compound (BMC) prepregs, and glass mat thermoplastics (GMT). SMC and BMC are applicable to thermoset matrices, whereas GMT is normally used for thermoplastic matrices. CM can also be divided into two types, namely cold and hot CM. In cold CM, only pressure is applied, as it requires only room temperature for the curing process, whereas both heat and pressure are necessary for the latter [32]. The mould is preheated before transferring the heat to the composite and starting the curing process. Plastic materials are placed in between two preheated moulds, which are then pressed against each other and take the shape of the mould cavity with great dimensional accuracy. The process is carried out at a high temperature and pressure, depending on the requirements of the composite, for a set period until the moulding material is shaped [29,33]. It is also critical to keep the pressing time under control. Otherwise, there is a risk of cracking, scorching, or warping [28]. The advantages of the CM include a short cycle time and the potential to mould large, complex parts in a variety of forms and sizes. It helps to reduce waste material, which gives it a significant benefit when working with high-cost materials. In addition, it is one of the least expensive moulding techniques when compared to other production processes such as IM and RTM [31].

In extrusion, a thermoplastic resin is heated and plasticised through the action of the barrel of the extruder and the rotating screws. It is then extruded and driven out of the chamber via a die to form different cross-section products. This method has been adapted to fabricate short-fibre composites and may be utilised to make an IM precursor [30]. There are two types of extruder machines: single-screw and twin-screw. The former provides consistent mixing and distribution depending on the material’s viscosity, whereas the latter is mostly preferred with short fibres to achieve more intense mixing [34]. A twin-screw system is applicable in the field of compounding and polymer blending, which gives better mechanical performance than a single-screw extruder [26]. Thermoforming is a unique method of transforming thermoplastic plastic sheets into functional plastic products. The sheet material is clamped and heated until it softens. Under the action of an external force (vacuum or air pressure), the softened sheet is forced against the shape of the mould. After cooling and shaping, the product is finished. Food packaging is the most prevalent use for thermoformed containers, trays, cups, and jars [11,35].

## 3. Mechanical Properties of Different Types of Bioplastics

There are various mechanical assessments performed to comprehend the composite with indicators including tensile, flexural, impact, and hardness tests. The study of mechanical characterisations from different tests reveals the mechanical behaviour of a polymer composite and provides information on the composite’s suitability for its intended purpose [7,36]. The design criteria and specific application of the composite may be accessed by comparing multiple properties [37].

### 3.1. Tensile Properties

Tensile testing is among the most basic and intensively studied mechanical testing methods for polymeric materials, owing to its simplicity of testing and ease of interpretation of the results. It is often termed “tension testing” and is used to evaluate the stress-strain behaviour under tension [7]. During tensile testing, a tensile force (pulling force) is exerted on the material, and the specimens’ response to the applied force (stress) is quantified. The samples are subjected to controlled tension until failure occurs. A stress-strain curve can be developed through this test to obtain the tensile strength, modulus of elasticity (Young’s modulus), and elongation at break of the samples assessed [38]. The tensile strength indicates the highest stress that the specimen could sustain before it broke, as well as how much it stretches before it breaks [39].

### 3.2. Flexural Properties

The flexural test, also called the transverse beam test, is used to measure flexural strength and flexural modulus. It is a basic parameter to determine the feasibility of composite materials for structural applications [36]. Flexural properties are the outcome of the simultaneous effect of tensile, compressive, and shear stresses in the materials. Under flexural loading, a rectangular cross-section specimen is loaded in either a three-point bending or four-point bending mode [7]. The interfacial connection between fibre and matrix, and the extent of tension transfer between fibre and matrix, governed the flexural properties [5]. The flexural strength indicates the maximum breaking stress at failure, whereas the flexural modulus reflects the ratio of applied stress to deflection, computed from the starting slope of the stress-strain deflection curve [7,39,40]. These two values tell us the sample’s resistance to flexure or bending forces.

### 3.3. Impact Properties

The impact test is used to evaluate structural materials’ impact strength, toughness, and notch sensitivity, measured either by Charpy or Izod impact tests. The impact strength reflected the material’s capacity to tolerate high-rate loading [7,39]. Toughness is a measure of the total energy absorbed per unit volume of material up to the point of rupture. Its value is quantified by measuring the total area under the stress-strain curve. Tough and brittle materials have high and low work-to-fracture values, respectively. Most polymer materials require an impact test because it concerns product performance, safety, liability, and service life [21,36,41].

### 3.4. Hardness

The hardness of a material signifies how resistant it is to abrasion, indentation, scratching, and plastic deformation under compressive load. These properties outline the material’s wear and tear [42]. The surface hardness value can be determined by the Rockwell, Vickers, Brinell, and Shore hardness testers [43,44]. A greater hardness value suggests that the material is tougher and more resistant to penetration by other materials. From an engineering standpoint, a material’s hardness is a benefit because it makes it less likely to wear down due to friction or erosion by water, oil, or steam [36].

## 4. Reinforcement Materials for Bioplastics

Bioplastics in the biocomposite category can be reinforced with reinforcing agents like inorganic fillers and natural fibres to increase their mechanical strength [1,3]. Fillers are often inexpensive, making the filler-reinforced biocomposites more economical. Importantly, these biocomposites unlock a new channel for the creation of innovative polymeric materials with more fascinating features [45]. In composite materials, a polymer matrix holds the fibres together, transfers the load to them, and distributes the load evenly. The fibres, on the other hand, serve as the primary load-bearing component because of their superior strength and modulus [46]. Figure 1 presents a schematic diagram of the filler–matrix interface, in which stress acting on the matrix is conveyed to the filler via the interface. For the composite to perform effectively, both phases must be well-bonded. When stress is transmitted, fibre–matrix adhesion is localised to the interphase region, which is a three-dimensional area between the phases, as illustrated in Figure 1b. With the formation of a “bridge” in the interface region, the efficiency of stress transmission can be enhanced. Likewise, the structure and properties of the interface are considered since they have a direct influence on the physical and mechanical characteristics of composite materials [47].

### 4.1. Inorganic Filler

Synthetic man-made fibres created by chemical synthesis are further categorised as organic or inorganic based on their composition. Glass fibre (GF), carbon fibre (CF), metal fibre, and ceramic fibre belong to the category of inorganic synthetic fibres [48]. These fibres are made from materials such as silica, alumina, aluminium silicate, zirconia, boron, boron carbide, boron nitride, graphite, silicon boride, silicon carbide (SiC), silicon nitride (SiN), etc. Inorganic fibre composites have been used to make lightweight structural materials with excellent strength and modulus values that can be tuned to specific loading requirements. They are heat-resistant, more robust, stiff, and have a higher melting point than conventional fibres [49].

Among these fibres, glass and carbon fibre are the most commonly used, with significant advances in plastic reinforcement applications. GF was the first continuous inorganic fibre, invented around 80 years ago [50]. They are abundant, available at cheap rates, and easy to use. GFs come in a variety of structures, including rambling, chopped strands, threads, fabrics, and mats, serving distinct functions for GFRP composite formation [31]. Numerous types of GFs are commonly used in polymer composites, depending on the raw materials employed and their quantities. Examples of GFs are A-Glass, C-Glass, D-Glass, E-Glass, R-Glass, etc. E-glass is the most universally used fibre in polymer composites [49,51,52]. GFs are usually employed in electronics, marine, aviation, and automotive applications. They have superior strength and durability, as well as thermal stability, impact resistance, electric insulators, and incombustibility.

Carbon fibres are fibres containing at least 92% carbon by weight, which first emerged in the market in the 1960s. The outstanding mechanical strength, stiffness, modulus of elasticity, high-temperature tolerance, and chemically inertness accurately describe these fibres [49]. Furthermore, alkaline materials and ultraviolet (UV) light do not affect CFs. These properties have made them very appealing in numerous engineering industries, including aerospace, civil engineering, sports, marine transportation, and the automobile industry. The main weakness of CFs is their high capital cost when compared to GFs, plastic fibres, or naturally occurring fibres. Moreover, their impact properties are comparatively weaker than those of GFs, but they are stiff and strong like steel [31].

### 4.2. Lignocellulosic Materials

The use of lignocellulosic fibre reinforcement has dominated research in recent years. A plant-derived natural fibre is known as lignocellulosic fibre (LCF). These are composed of cellulose, hemicellulose, lignin, pectin, waxes, and other water-soluble substances. The composition and percentages of these components differ depending on the type of biomass. Hence, they exhibit diverse mechanical behaviour [46]. Given their abundance, low density, non-abrasive, high specific mechanical strength, and modulus, these fibres are suitable for composite materials where the ideal property is centrally weight reduction. In addition to the enhancement of the properties of biopolymer composites, LCFs have also been reported to improve biodegradation at the end-use [53]. For modern uses, LCF-reinforced composites are better than synthetic fibre-reinforced composites for two reasons: they are cheaper and better for the environment.

The major framework component of lignocellulosic biomass is cellulose, which is a long-chain polysaccharide composed of D-glucopyranose units interlinked with β-1,4-glucosidic bonds [54]. Cellulose exists in both crystalline and amorphous regions, with crystalline cellulose consisting of chains with an orderly molecular arrangement and amorphous cellulose consisting of random arrangements [55]. Crystalline cellulose imparts strength and stability to the fibre. Because of the strong intramolecular hydrogen bonding, the hydroxyl groups (OH) within crystalline cellulose molecules are impermeable to chemicals, even water molecules. On the other hand, amorphous cellulose is soluble and more susceptible to enzyme degradation. The OH group in this region forms hydrogen bonds, allows water molecules to pass through, and gives polar fibres [46,56].

In contrast to cellulose, hemicellulose is a short-branched heteropolymer that exists in plants as an amorphous form, connected to cellulose microfibrils via hydrogen bonding, providing structural support to the fibre [57]. Hydrophilic hemicellulose is more prone to alkali and acid degradation [27]. Structurally, amorphous lignin synthesised by phenylpropane units that are arranged in a complex three-dimensional network structure is non-water-soluble and optically inactive. Lignin, coupled with cellulose and hemicellulose, provides additional strength to the hemicellulose–cellulose network [58,59].

### 4.3. Bacterial Cellulose

In addition to plant-based biomass, bacterial cellulose has become in demand due to its purity (without lignin, hemicellulose, and pectin as compared to plant cellulose), high aspect ratio, and high crystallinity [60]. Owing to its non-toxicity and mechanical stability characteristics, bacterial cellulose has received high demand for biomedical medical applications [61]. Wang et al. [62] have proven that bacterial cellulose could be used as a reinforcement to improve the mechanical properties of the composites as compared to the neat thermoplastic matrix. This improvement could be further enhanced when the bacterial cellulose was esterified, where a better bacterial cellulose distribution was indicated due to better interface compatibility [63]. Hence, based on the performance of bacterial cellulose-reinforced thermoplastic, it is believed to be an effective reinforcement for bioplastic.

## 5. Mechanical Properties of Reinforced Bioplastics

In the past decades, it has been a research hotspot to optimise the characteristics of polymers by integrating fillers, and research articles have been published to update the forefront of polymer-based composites for structural applications [4,37]. In this section, the focus is given to the mechanical behaviour of biocomposite materials. These mechanical studies are pivotal factors in correlating the changes to the bioplastic following reinforcement by fillers since their outcomes are highly related to the distribution and orientation of fillers within the matrix [28]. Figure 2 depicts the effect of the filler distribution in the composite system. The presence of uniformly distributed filler in the matrix allows the load to pass through without cracking or delamination. In contrast, poor filler dispersion and agglomeration create stress concentration points within the composite (Figure 2b). When the composite is loaded, cracks may initiate and propagate, resulting in a reduced load-bearing capacity in the fractured composites [46]. As previously stated, it is primarily driven by filler–matrix adhesion. The stress load can be effectively conveyed across the interface by strongly bound particles. In reverse, filler–matrix debonding causes physical discontinuities that cannot withstand mechanical forces [40,64]. The homogeneity of the filler in a composite system is also dependent on the mixing technique employed. The most popular mixing technique for preparing a reinforced thermoplastic composite is extrusion using a twin-screw extruder. The process parameters of a twin-screw extruder, i.e., the design of the screw and the co-rotating mechanism, determine the homogeneity level of a thermoplastic mixture [11,27]. Hence, the filler agglomeration issue can be resolved if a good mixing technique is used in which the stress concentration point formation can be avoided and the distribution of the load can be dispersed well, as depicted in Figure 2a.

In the composite system, the interaction between the thermoplastic and the filler can be classified into physical and chemical interactions. Mechanical interlocking is a form of physical force that holds two components of different interfaces together [65], whereas the formation of chemical bonding via functional groups between filler particles and matrix is classified as chemical interaction [43]. Without physical interaction, the filler may slip from the matrix when loaded, resulting in reduced stress transfer efficiency and lower composite strength. The composite strength can be further enhanced with the formation of chemical linkages between the filler and the matrix. For example, the functional group of isocyanates in the polyurethane matrix interacts with the OH groups of lignocellulosic filler for urethane linkage formation. This linkage may serve as a bridge where the load may be transferred efficiently from the matrix to the filler. With this, the strength of a composite can be enhanced [66].

Jiménez et al. [67] created a biocomposite using natural fibre reinforcement from sugarcane bagasse (SB) and a biodegradable starch-based matrix, Mater-bi^®^ (PTA). SB fibres are prepared in the form of sawdust (WF), mechanical (MP), thermomechanical (TMP), and chemical-thermomechanical (CTMP) pulps. After incorporating 30% *w*/*w* of bagasse into the matrix, it turned out that the PTA/WF composite had a lower tensile strength than the neat PTA matrix. The major explanation for the decrease in tensile strength is that sawdust particles with a low aspect ratio are not perfectly adhered to the PTA matrix and have reduced reinforcing capabilities. The particles acted more as filler than as reinforcement. Conversely, all of the remaining fibres with higher aspect ratios behaved as reinforcements and produced composites with higher tensile strength. Moustafa et al. [53] identified the effect of reinforcement on the mechanical properties of the resulting composites. They incorporated coffee grounds (CG) into the PBAT matrix at varying levels of content, ranging from 10% to 50% in the presence or absence of PEG plasticiser. For PBAT/CG composites without PEG, large cavities are found in the rough fracture surface, translating into poor interfacial bonding between CG particles and the matrix. The tensile stress–strain curve demonstrated a significant loss in the mechanical properties as filler loading increased. In contrast, the PBAT/CG50 biocomposites had the highest elastic modulus values (777 MPa) of all the samples. However, the plasticization effect on the same PBAT/CG50 biocomposites made the values drop to 111 MPa.

Totaro et al. [68] highlighted the excellent mechanical properties of bioplastics by creating composites with silver skin coffee (SSK) as a filler and PLA and PBS as matrices. The incorporation of coffee by-product increased the value of the elastic modulus for both polymers, confirming optimum dispersion and wettability of the filler in the corresponding matrix. Aydemir and Gardner [24] explored the influence of cellulose nanofibrils (CNF) on the mechanical properties and discovered that the presence of CNF provides a mechanical property enhancement of 4–18% in the PHB/PLA blends. The introduction of filler to starch-based systems was investigated by Collazo-Bigliardi et al. [19]. They prepared coffee and rice husk cellulose fibres reinforced glycerol plasticised TPS films at 1, 5, and 10 wt%. For all composites, including both fillers, there was a strong tendency for the stiffness of the samples to increase. The obtained elastic modulus value reflects that even 1 wt% of filler enhanced the modulus of composites by approximately 60%, independent of the fibre type. Such an improvement might be attributed to the high purity of cellulose in the generated fibres, reflecting a higher degree of crystallinity of the material, which benefited the bonding between fibres and starch. Interestingly, adding 1 wt% of coffee fibre to TPS film did not change its ability to stretch, but adding the remaining 5 wt% and 10 wt% of both fillers made composites that were less stretchy.

Baek et al. [69] discovered that incorporating fibres into polymer matrices generates unstable interfaces and that the fibre’s positive reinforcing effect is underutilised. The tensile strength of the green composites with PLA is negatively affected by the addition of natural fillers like bamboo flour (BF) and CG due to weak interface and obstructing stress concentration. A similar polymer matrix and filler combination were used by Kumar and Tumu [70], incorporating 5 wt% bamboo powder (BP) into the PLA matrix. When compared to neat PLA, the tensile properties of PLA/BP composites were reduced because of inadequate bamboo fibre dispersion and poor interfacial compatibility between the reinforced BP and matrix. When the adhesion is weak, the fibre does not play its role as a reinforcing material. Based on the data presented above, it can be assumed that the reinforcement of bamboo fillers weakens the mechanical properties of PLA/bamboo filler composites.

## 6. Factors Affecting the Mechanical Properties of Reinforced Bioplastics

The structure and final properties of composite materials, notably their mechanical properties, are dependent upon the reinforcement and the polymer matrix, and the interaction between the two constituents [26,71]. The type of filler, aspect ratio, filler loading, orientation, and many more are all important considerations. All these factors can be tuned to yield an optimum combination of mechanical strength and stiffness for future applications [52]. The contributions of surface treatments to the tensile, flexural, and impact properties of composites made from reinforced fibres will be discussed in the rest of this paper.

### 6.1. Types of Fillers

Strengthening polymers with fibres in varying ratios opens a world of options for developing materials with different attributes. These benefits have inevitably resulted in its widespread use in polymer applications, as it is observed that final properties are not feasible with a single polymer alone [43]. Fillers are widely used to improve the processability and mechanical properties of polymeric materials while at the same time lowering the material costs [47]. For reinforced composites, the dominant factor that governs their properties is the filler type and its properties. The behaviour of fillers should be studied to understand their actual contribution to the composite before incorporating them into the composite material [5,58]. Along with that, high-performance composite applications can only be met if there is a homogeneous dispersion of reinforcement inside the matrix and proper bonding between them to allow appropriate stress transmission from fibre to the matrix and vice versa. In other words, both components must be physically and chemically compatible [7,72,73]. In LCFs, mechanical properties vary subject to the composition and structure of fibres, which are listed as the following factors: fibre diameter, spiral angle of fibrils, degree of crystallinity, size of crystalline fibres and non-crystalline region, and chain orientation [7]. Moreover, the chemical composition of fibre represented by the percentage of cellulose, hemicellulose, lignin, and wax differs from plant to plant and throughout distinct areas of the same plant. Cellulose showed higher stiffening abilities than hemicellulose, while lignin is typically used as a coupling bonding agent between cellulose and hemicellulose. Therefore, the high cellulose content in the fibre explains the increase in mechanical strength [59]. Other factors to consider are the plant’s growing conditions, such as its topography, climate, and age [46,74].

### 6.2. Aspect Ratio

The aspect ratio of fibre is a valid indicator of the reinforcing abilities of a certain fibre [11]. It is the length-to-diameter ratio of a fibre. This ratio depends on the extrusion process that breaks or shortens the fibre bundles. In general, when the aspect ratio increases, the stress may be transferred more effectively since there are more surface areas available for interaction. Interfaces are ruled by interfacial bonding, which is a key issue in composite science because it determines how stress can be transferred between the matrix and the fibres, compromising the mechanical characteristics of the whole material [34].

In a short-fibre composite, the tensile load applied is transferred to the fibre through shear loading at the interface. The tensile stress is zero at the fibre ends and increases along with the fibre length. Therefore, the ideal fibre length should be larger than the critical length (Lc) in order to properly convey the load during tensile loading [26]. When the fibre length is less than the critical length, debonding and pull-out of fibres will occur, indicating poor interfacial bonding in fibre-reinforced composite systems [75]. The value of the critical length can be determined as follows:$$Lc=\sigma_{f}D/2\tau$$
where $\sigma_{f}$ is the tensile strength of fibre, D is the diameter of the fibre, and τ is the fibre/matrix interfacial shear strength (IFSS) [7].

It is worth emphasising that the real reinforcing ability of fibre corresponds to the aspect ratio. Fibre with aspect ratios greater than 10 behaves as reinforcements, allowing preferential dispersion and distribution, which positively influences the mechanical behaviour due to particle alignment. On the other hand, lower aspect ratio fibres have less reinforcing capability and can even cause mechanical failure [67]. In the study of García-García et al. [40], the addition of spent coffee ground (SCG) particles (aspect ratios lower than 2) resulted in a deterioration in flexural strength. This is because no particle alignment can be accomplished during manufacturing. Karaduman et al. [76] reported that enzymatically treated jute fibre-reinforced polyester composites had a decrease in fibre diameter, which increases the aspect ratio. It created a large effective contact surface for resin impregnation and achieved great relevance to the final properties of the composites. When the fillers are compatible and have the right aspect ratio, they can intensify the material and allow it to be used in composites [5,11].

### 6.3. Percentage of Filler

The relative proportions of the filler materials in the formulation dictate the mechanical properties of the composites [68,77]. Various studies were undertaken on the effects of filler on the mechanical properties of biocomposites with different filler contents, and it can be said that aggregation phenomena are more evident for specimens at high filler loading [24]. In the study of calcium phosphate (CaP) and magnesium phosphate (MgP) nanoparticles’ impact on pure PLA by Sahu et al. [77], the tensile strength of PLA nanocomposites increases linearly with the inclusion of CaP concentration up to 15 wt%, with a subsequence decrease in tensile strength at 20 wt% of fibre concentration. According to their team, this increase in tensile strength is related to the presence of a tensile stress-carrying filler in the polymer matrix. Similarly, the tensile strength of PLA/MgP composites confirms that beyond 2% MgP concentration, minimal improvement is observed. This behaviour is attributed to the presence of agglomerates on the surface at a concentration of 20% CaP nanoparticles and above 2% concentration of MgP nanoparticles, as observed by scanning electron micrographs. Ragoubi et al. [78] discovered that Young’s modulus and deformation at break of PLA/miscanthus composites decreased at a higher ratio of filler. Indeed, signs of aggregation are present and heterogeneous dispersion of filler in the polymer matrix occurs, precluding the transmittance of stress from fibre to matrix. The results of X-ray tomography also showed that at this ratio, composites have larger holes and higher porosity. Due to the intrinsic stiffness of miscanthus fibres, the increment in reinforcing content reduces deformation at break significantly. The material becomes less plastic. This weak structure of the blend reversed the mechanical strength of PLA/miscanthus composites. Using coffee grounds and PBAT composites, Moustafa et al. [53] obtained equivalent results.

In summary, the mechanical properties of fibre-reinforced composites are found to improve linearly with increasing fibre content up to a certain optimum value. Further addition of fibre above that limit adversely affects the mechanical strength due to increased porosity and the formation of agglomerates. Likewise, there is insufficient resin to properly wet all of the fibres, preventing good filler dispersion within the matrix and limiting the two from sharing stress. Such an effect can be related to the degradation of mechanical properties [7]. Hence, future research should concentrate on the effect of filler loadings on the mechanical properties of composites, as composites with uniformly distributed particles yield a higher load capacity.

### 6.4. Types of Treatment for Reinforcement Materials

Researchers encountered several problems when fabricating biocomposites reinforced with natural fibres, most notably the hydrophilic nature of natural fibres, thermal instability of natural fibres, and a weak interface between the reinforcing phase and matrix phase, particularly for a matrix with hydrophobic behaviour [27,44,58,79]. Fibres and polymer matrices have distinct chemical structures. Fibre dispersion is dependent on compatibility [80]. The compatibility issues caused by the hydrophilic fibre and hydrophobic polymer matrix restrict the future application of composites, especially in an outdoor environment [7]. This is because the OH groups in the amorphous region of the fibre create new hydrogen bonds with water molecules from the atmosphere, resulting in extremely high moisture absorption. Consequently, fibre swells within the matrix, creating a poor linkage to the matrix that drives stress concentrated at the interface. In addition, micro-cracking also occurs between the swollen fibres and matrix, leading to dimensional instability with a detrimental effect on the mechanical performance of the resulting composite [7,51,81]. This confirms the need to remove hydrophilic OH groups and surface particles from the fibre surface via the surface modification process. To circumvent the heterogeneous interfacial problem, various fibre surface treatments have been proposed, namely physical, chemical, and biological treatments.

#### 6.4.1. Physical Treatment

Physical treatment is aimed at increasing the mechanical bonding of the polymers by modifying the fibres’ structural and surface properties without changing the chemical composition of the fibre extensively [7,51]. In other words, a stronger mechanical connection between the fibre and the matrix typically improves the interface. Physical approaches include mechanical comminution (chipping, grinding, milling) and electric discharge (plasma, corona, ultraviolet (UV) radiation, electron radiation) [6,58]. The objective of chipping and grinding is to disperse the particle size and facilitate the treatment process. This process is followed by milling methods, which can be ball milling, two-roll milling, hammer milling, etc., to become fine powder [54]. In the end, the crystallinity of the fibre is reduced, which affects the degree of polymerisation. The hydrolysis rate and mass transfer characteristics can be improved due to the reduction in crystallinity and particle size, respectively. The grinding conditions and intensity influence the final particle size of fibres, which in turn determines the energy requirement for mechanical comminution. This implies that mechanical processes are energy-intensive to achieve a high fermentable sugar yield, which is not economically feasible [82,83].

Plasma treatment is considered an environmentally friendly method for surface treatment using no chemical solvent. The plasma flows modified the fibre surface through ablation, etching, crosslinking, and surface activation. The fibre surface strength is enhanced after crosslinking the surface with free radicals [84,85]. Plasma etching generates hydrophobic surfaces by providing the desired roughness to the surface for physical adhesion and introducing new functional groups for higher polarity fibre surfaces [73,79]. The functional groups establish strong covalent bonds with the matrix, generating surface crosslinking to boost surface energy. In the end, the crosslinking process contributed substantially to an increase in mechanical strength [11]. The surface hydrophobicity can be altered by adopting different plasma parameters of exposure, i.e., nature of gas, exposure time, and applied power [85].

Another method of atmospheric plasma technique is corona treatment, which uses electric current to transmit changes in fibre properties and surface energy. Using corona discharge, chemical (surface oxidation) and physical (etching) effects are generated on treated fibres. Air plasma species bombardment increases surface roughness and coarseness, contributing to improved interactions between fillers and matrices [78,86]. To summarise, physical treatment is a non-polluting process with a short processing time and no specific conditions. During the treatment, a huge amount of material may be applied on a large scale, which benefits the manufacturing production of PF [58]. The fibre surface can be modified without affecting its integrity [79].

#### 6.4.2. Chemical Treatment

Chemical treatment, which alters the chemical composition, surface topography, and morphology of natural fibres, is the most widely used method for strengthening fibre-matrix adhesion [5]. This treatment is described as the formation of a covalent bond between some reactive constituents of LCFs and chemical reagents, with or without the use of a catalyst [7]. The integration of hydrophilic fibre and hydrophobic matrix induces fibre swelling within the matrix and weakens bonding strength at the interface [56]. Chemical modifications destroy the fibre’s hydroxyl groups and substitute them with hydrophobic chemical bonding. The seduction in the water absorption capability of the fibre is caused by the degradation of the OH group. In this context, fibre with lower hydrophilicity and a matrix with reduced cracking benefit the overall mechanical properties [87].

Acidic or alkaline treatments are the most commonly used and easiest treatments. These treatments usually focus on the fibre surface, where the soluble contents in fibres are dissolved using an acid (HCL) or alkali (NaOH) solution for hours, “washing” the surface from an uneven distribution layer of non-cellulosic components (lignin, hemicellulose, pectin, and impurities) that cover the fibre surface. These components are undesirable and may hinder the formation of physical, chemical, or both linkages between the matrix and the fibre (Figure 3). Alkali-treated fibres increased the ratio of exposed cellulose and experienced mass loss due to partial or complete elimination of non-cellulosic components, the majority of which are amorphous [45]. An easier fibrillation process is promoted after separating the fibre bundles into finer fibrils using a chemical solution to provide a larger surface area for interaction with the matrix [88,89]. Without impurities, fibre surfaces become rougher, providing additional sites for the polymer to anchor. Finally, potential mechanical anchorage and extra load-bearing capabilities at the interface can be accomplished as the fibres are surrounded by the matrix [79,90].

Coupling agents like silane, maleic anhydride, permanganates, and acetic acid function as the bonding agents to “bridge” the hydrophilic fibre and hydrophobic polymer through covalent bonding, hydrogen bonding, or polymer chain entanglement [90]. Chemically treated fibre has high moisture resistance properties via the removal of an OH group coating on the fibre surface. Grafting with compatibilisers (maleic anhydride) (MA) is a useful approach that allows the functional surface of fibre and matrix to interact efficiently. MA connects with OH groups in the fibre via covalent bonding and removes them from the fibre. The hydrophilic nature is reduced after a long polymer chain coating on the fibre surface. A maleated coupling agent creates a carbon–carbon connection between the OH groups of the fibre and the anhydride groups of MA. This covalent bond makes a bridge interface for efficient interlocking [43,56]. Silane treatment involves hydrolysis of alkoxy groups on silane to form silanol (Si-OH). During the condensation process, one end of silanol interacts with the OH group of cellulose (Si-O-cellulose), while the other end interacts with the functional group in a matrix (Si-matrix), forming a siloxane bridge between them. The number of OH groups of cellulose is reduced in the fibre cells, increasing the surface’s hydrophobicity and ameliorating the interface’s strength [6,26,91]. Acetylation substitutes the OH groups in fibres with acetyl groups, rendering the fibre surface more hydrophobic and rougher, providing stability to the composites [59]. Certainly, chemically treating fibre has significant benefits. However, there are some drawbacks. The well-recognised weakness is that these treatments provoke environmental issues attributable to the use of hazardous chemicals, inappropriate handling of chemical waste, and the generation of difficult-to-dispose-of by-products. This issue adds extra cost to the production process, making this treatment less widely adopted in manufacturing inexpensive products [44,92].

#### 6.4.3. Biological Treatment

Given the environmental benefits, there has been an increasing interest in biological treatment. This treatment makes use of biological agents, either the microorganisms or enzymes secreted by the microorganisms, to fragment complex molecules of biomass into their constituents and change the structure and chemical composition of the fibre so that the treated fibre is more amenable to enzyme digestion [76]. Generally, this treatment is performed using different fungal species like white-, brown-, or soft-rot fungi and bacteria. White and soft-rot fungi specifically focus on both lignin and cellulose, whereas brown rot fungi depolymerise cellulose and hemicellulose. The specific extracellular enzymes secreted by these microorganisms increase the rate of enzymatic hydrolysis of the substrate through lignin degradation. White-rot fungi are reported to be the most efficient among these microorganisms, with *Phanerochaete chrysosporium* serving as the model strain for lignin breakdown [93]. Figure 4 illustrates the possible mechanical interlocking between filler and matrix in a composite system. The waxy layer coating the external surface is primarily responsible for the smooth native fibre surface [87]. Fungi produce hyphae during treatment, which create fine holes (pits) on the surface and provide roughness to the interface. It is believed that some of the filler components could be removed. A rougher fibre surface provides additional anchoring points, increasing the likelihood of mechanically interlocking with the matrix. Consequently, a high level of filler/matrix adhesion and good mechanical behaviour of the composite compared to the one with a smoother surface are expected [44,91,92,94].

Microorganisms are accountable for lignocellulosic materials degradation and demineralisation owing to the production of two types of extracellular enzymatic systems: the oxidative ligninolytic system, which acts on the phenyl rings in lignin, and the hydrolytic system, which attacks the cellulose and hemicellulose to liberate fermentable sugars [82]. Three major enzymes participate in the oxidative ligninolytic system: lignin peroxidase (LiP), manganese peroxidase (MnP), and laccase. The H_2_O_2_-dependent oxidation of lignin is catalysed by the LiP and MnP enzymes, while the demethylation of lignin components is catalysed by laccase, a copper-containing enzyme [93]. However, not all of these enzymes are secreted by fungal cultures. Bacterial laccase has also been identified in *Azospirillum lipoferum*, *Bacillus subtilis*, and other organisms, but they are thought to have a minimal lignin degradation capability. Other than fungal treatment methods, several industrial enzymes, such as xylanase, cellulase, laccase, and pectinase, play a vital role in enzymatic hydrolysis. Non-cellulosic compounds conceal the external fibre surface and develop poor surface wetting, which impacts the interfacial adhesion between fibre and matrix [5,95]. Therefore, the enzyme catalyses biochemical reactions by binding a substrate at the active site specifically. Xylanase breaks down the hemicellulose, cellulase removes cellulose, laccase degrades the lignin structure, and pectinase is responsible for pectin degradation [76,96]. The modified fibre is less hydrophilic and has more exposed cellulosic fibrils, which improves wettability and mechanical interlocking between the fibre and matrix [87,97].

Unlike physical and chemical treatment methods, biological treatment involves mild operating conditions and lower energy input and does not require acids, alkalis, or any reactive species. This process does not generate inhibitory substances or undesirable products. Another reason for embracing biological treatment is its cost-effectiveness, as no chemicals are employed and post-treatment washing and/or detoxification are not required [80,95]. However, there is a need to monitor the growth of microorganisms regularly and prolong treatment time for effective delignification, which restricts the industrial application of this method. Thus, the key parameters, such as the type of microorganisms involved, inoculum concentration, treatment time, pH, and temperature, should have optimum values to improve the techno-economic performance of the biological treatment. Despite that, biologically treating natural fibres is an innovative and emerging trend, considering the enzymes used are readily accessible and economical, and because the enzymes can be recycled, they produce little or no waste [71].

## 7. Effect of Filler Treatment on the Mechanical Properties

Prior to their incorporation into polymeric matrices, the morphological characteristics of fibre surfaces should be modified to minimise the shortcoming that comprises weak fibre-matrix interfacial attachment [91]. As previously discussed, the primary goal of surface treatment of filler is to achieve a high degree of fibre-matrix interlocking and stress transferability of the composites. Overall, the fibre surface treatment affects the physical, chemical, thermal, and mechanical properties of fibres and the resulting composite. The characterisation of mechanical properties, such as tensile strength, flexural strength, impact strength, and interfacial shear strength is studied to find out the effectiveness of various surface treatments on the performance of the resulting composite materials. The biocomposite after the application of different surface treatments to the fillers is summarised in Table 1.

### 7.1. Physical Treatment

Natural fibres have been physically modified to promote fibre-resin adhesion in fibre-reinforced composites, including plasma, corona discharge, and electron beam treatments [58]. Physical modification raises the IFSS of neat fibres with the polymer matrix, which was previously low before surface treatment. Gibeop et al. [79] studied the mechanical properties of jute fibre/PLA biocomposites by treating them with helium and acrylic acid as carrier gas and monomer, respectively, with a plasma power of 3 kV and 20 kHz for different exposure times (30, 60, and 120 s). Plasma polymerised fibre composites outperformed alkali-treated composites in terms of tensile strength, Young’s modulus, and flexural strength by up to 28, 17, and 20%, respectively. Added to that, plasma-treated jute fibre composites aided in a pronounced improvement in the IFSS, which was determined by a micro-droplet test. The increment in IFSS value of 90% more than the untreated jute fibre/PLA composite was attributed to a rougher fibre surface indicated by an increased surface friction coefficient value. This is subjected to the heat and etching effect on the outer layer of fibre, which leaves more non-polar lignin on the surface. These results provide a great contact between jute fibres and the PLA matrix, which could be visualised by SEM micrographs. The significant improvement in the mechanical performance of the resulting composites suggests that plasma surface modification is capable of increasing the connectivity between hydrophobic matrix and hydrophilic fibre.

In a study performed on plasma treatment, de Farias et al. [73] treated coir fibres with oxygen and air before incorporating them into the TPS matrix. Their study demonstrated that plasma treatment (80 W, 7.2 min) was effective in improving both the tensile strength and elastic modulus of the composites when either oxygen or air was used. When compared to air plasma, oxygen plasma was more influential in all conditions, with the composite’s tensile strength and elastic modulus achieved by up to 300% and 2000%, respectively. Stronger oxygen plasma etching removed more surface lignin, exposed the crystalline cellulose, and increased surface roughness and compatibility factor. The roughened surface points to fibre–matrix interlocking, which has a pronounced effect on the load transfer between them. The authors also pointed out that there was a correlation between plasma power and the resulting properties of the composites. Given this, these variables should be chosen wisely to reap the benefits of plasma treatment.

Miscanthus fibre was subjected to corona treatment at a discharge frequency of 50 Hz and a voltage of 15 kV for 15 min [78]. The fibres were blended with PLA granules containing 20–40 wt% fibre content, and the mixture was then extrusion-compression moulded to produce PLA/miscanthus composites. They experimented with both untreated and corona-treated fibres. Tensile measurements were used to determine the effect of fibres on the mechanical properties of PLA and composites. The effectiveness of corona-treated miscanthus fibres can be seen in the improvement of mechanical properties, including elastic modulus, stress, and strength at yield, in resultant composites when compared to PLA and composites containing unmodified fibre. Low fibre content (20% and 30%) showed better enhancement in Young’s modulus than the higher one (40%) because good fibre dispersion is conducive to better stress transmission from matrix to fibre. The chemical (surface oxidation) and physical (etching) effects of corona treatment on fibres could explain the improvement in interfacial compatibility between PLA matrix and miscanthus fibres, observed using X-ray photoelectron spectroscopy (XPS) and SEM. At higher ratios of treated fibre, the composites display larger voids and higher porosity, while Young’s modulus remained unchanged compared to composites with non-treated fibres. Amirou et al. [86] conducted another corona discharge treatment on date palm fibre (DPF) and PLA using the same corona discharge frequency and treatment time as the previous author. Extrusion-compression moulding techniques were used to create fibre mixtures with varying fibre content ranging from 30–40%. Before treatment, the inclusion of DPF did not show any improvement in the tensile strength, indicating inadequate adhesion between fibres and the PLA matrix. Through the corona treatment, there was a considerable improvement in tensile strength and Young’s modulus, with the highest elastic modulus (2951 MPa) reached by 30% reinforcement of palm fibres in polylactic acid compared to untreated reinforcements (2708 MPa) and the PLA matrix (2396 MPa). This is attributed to the mechanical anchorage related to an etching effect caused by the bombardment of the air plasma species on the fibre surface. Indeed, the specimen surface became rougher and coarser. In both studies, it was found that higher mechanical anchorage helped improve the interfacial contact and compatibility between the two phases.

Kumar and Tumu [70] have utilised electron beam (E-beam) irradiation at various doses (30, 60, and 90 kGy) to achieve better interfacial adhesion of BP and PLA. E-beam irradiated bamboo powder (EBP) was melt blended with PLA at 5 wt% and 10 wt% concentrations, as well as the coupling agent epoxide silane (3-Glycidoxypropyltrimethoxy silane) (ES). They have asserted that the PLA/EBP5/ES 5phr with 5 wt% EBP and 5phr ES has better tensile properties than other PLA/BP composites. This could be because trapped free radicals in the EBP initiated the interaction with carboxylic terminal groups of PLA and epoxide groups of epoxide silane, forming PLA-g-ES copolymers. Because the silane alkoxy groups of PLA-g-ES are extremely reactive to the hydroxyl groups of bamboo powder, the copolymers function as an interface between the PLA matrix and the fillers to improve their miscibility. Besides, the composites have shown a noticeable improvement of 12% in the notched impact strength compared to pure PLA and rougher morphology with ideal distortions, indicating more impact energy was absorbed. The author points out that the incorporation of a higher percentage of EBP (10 wt%) leads to a decrement in the tensile properties because interfacial compatibility between matrix and filler decreases at a higher bamboo fibre content. Heterogeneous phase morphology, as corroborated by the SEM micrographs, which reflect a lack of adhesion between matrix and filler, may have contributed to lower mechanical properties. They also studied the effects of irradiation dose and concluded that a high-dose electron beam will generate excess free radicals that disrupt the intermolecular hydrogen bonding among the cellulose molecules.

### 7.2. Chemical Treatment

Most previous research identified that alkali-treated fibre improved the mechanical properties of the resulting polymer composite [51,88,89]. Boonsuk et al. [88] performed alkali treatment on rice husk (RH) using a high alkali concentration (11 wt% NaOH) and added it to the thermoplastic cassava starch (TPS) matrix at loadings of 5–20 wt%. The mechanical properties of untreated and alkali-treated RH/TPS biocomposite were studied and compared. The findings revealed that the addition of 20 wt% alkaline-treated RH/TPS biocomposites gave the highest tensile strength by 220% compared to the neat TPS but decreased elongation at break. The rough surfaces of treated RH and loss of hemicellulose after NaOH treatment recorded improved interface interaction and more effective fibre-matrix load transmission. Alkali treatment creates a smoother inner surface, splits fibres into fibrils (fibrillation), and makes OH-rich fibrils more accessible. After hemicellulose and lignin are removed, new hydrogen bonds can form between cellulose chains. Thus, from the above-reported finding, it can be extrapolated that the composites with high fibre content resulted in better tensile strength. In another study, alkali-treated alfa fibres were employed as reinforcement in PLA resin, and composites were prepared using IM with a fibre content of 20 wt% [89]. When 20 wt% NaOH-treated alfa fibres were included, the composite’s tensile strength and Young’s modulus were strengthened by 17% and 45%, respectively. At the surface of the fibres, it was seen that the fibre bundles were opening up and the cementing components (hemicellulose, lignin, waxes, and oils) were disappearing. This made the surface rougher and caused a high degree of fibrillation.

Aside from alkali treatment, acetylation is a popular fibre treatment method. Fitch-Vargas et al. [98] investigated thermoplastics made from acetylated corn starch composites reinforced with acetylated sugarcane fibre (AcSF). The AcSF-reinforced starch-based composite was prepared by extrusion. Through chemical modification and interactions between fibre-matrix, mechanical interlocking between the two phases was improved, as evidenced by an improvement in mechanical properties with AcSF of up to 12%. The water affinity property was reduced by the presence of hydrophobic acetyl groups in the biocomposite. Nanni et al. [59] applied two types of fibre surface treatments on grape stalk (GS) powder. Acetylation and silanisation, which were later reinforced in the PBS matrix. Acetylation reduced the polarity of GS and made its rougher and spongier, increasing the possibility of mechanically interlocking with polymer chains during melt compounding. AcGS had the best mechanical performance of all the samples tested, with Young’s modulus increases from 616 MPa to 732 MPa. This trend is clarified by the degradation of hemicellulose under the harsh conditions of the acetylation process and is well interconnected between GS and the PBS matrix, supported by FTIR and SEM-FEG analysis. Moreover, acetylation worked well to minimise the moisture uptake of treated GS, showing that the surface of treated GS became less hydrophilic.

An investigation was carried out on the chemical treatments using (3-methacryloxypropyl) trimethoxysilane, MA, and NaOH on palm fibre (Macaíba fibre) (MF), which was subsequently melt extruded with polycaprolactone (PCL) [99]. Following that, the biocomposite with an MF concentration varying from 10–20% was then thermally, spectroscopically, mechanically, and morphologically characterised. For elastic modulus upon the addition of 10% treated fibre, silane treatment gave the best response among the treated samples and a neat PCL matrix, but NaOH treatment gave the lowest value, possibly due to excess delignification which weakens MF. Interestingly, biocomposites with 15% and 20% MA treated MF showed the highest elastic modulus among all the samples, most probably due to greater interaction between constituent components, namely PCL, fibre, and MA. Chemically treated biocomposites outperformed untreated ones in terms of flexural modulus. These enhancements are associated with enlarged contact points between fibre and matrix as a result of defibrillation. MA treatment also improved flexural modulus, which is thought to be related to the “anchoring” of succinic anhydride groups on the fibre surface and benefits the polar interaction between PCL and MF. Conversely, chemically treated MF biocomposites demonstrated lower impact strength than untreated MF biocomposites. This is owing to oil action in natural MF. The presence of oil in the pulp increases plasticization mechanisms, resulting in higher impact strength. The application of chemical treatment on MF and increased MF content lowers biocomposite elongation due to improved chemical interaction between MF and PCL, which restrains macromolecular movements, resulting in more stiff and brittle materials. Through the gathered findings in their work, the authors concluded that MA had the best mechanical performance and NaOH had the worst.

The creation of an interconnected network from silane treatment reduces the swelling property of fibre as a result of stable covalent bonds between fibre and matrix [104]. Lule and Kim [72] discussed coffee husk’s (CH) mechanical properties against silanisation with a silane agent, 3-Glycidoxypropyl trimethoxysilane (GPTMS). When 40 wt% silane-treated CH is reinforced in the PBAT matrix compared to the 40 wt% untreated CH-reinforced composite, mechanical parameters such as tensile strength, Young’s modulus, and elongation at break are significantly improved. SEM micrographs also showed continuous phase morphologies with no gaps between their interfaces, achieving good interfacial interactions with the polymer matrix, which promoted greater physical and mechanical characteristics of the composites. Figure 5 outlines the stress transfer efficiency between filler and matrix. The absence of a gap between filler and matrix is attributed to the possible interaction, such as the development of covalent bonds. As a result, the stress transfer efficiency from matrix to filler is expected to be higher than that without interaction (Figure 5b). The stress could not be transferred due to the gap between filler and matrix, as depicted in Figure 5a. This demonstrated that silane treatment aided stress transmission between CH and the PBAT matrix by preventing the formation of voids and gaps. The same author studied the incorporation of surface-treated silicon carbide (T-SiC) particles in PBAT and polycarbonate (PC) matrices, which led to a substantial enhancement in tensile strength and Young’s modulus, with a reasonable drop in ductility owing to greater SiC loadings [100]. According to Tanjung et al. [101], the inclusion of maleic acid-treated and silanated CS filler in the composite mixture has remarkably increased the PLA/CS composite’s tensile strength and Young’s modulus but reduced its elongation at break when compared to the untreated biocomposite. Wang et al. [105] studied the use of herb residue as a reinforcement material for PB. They found that the introduction of herb residue to PB improved its thermal stability, and this phenomenon was more obvious when the herb residue was treated with a silane coupling agent. This was attributed to the improvement of interfacial properties between the matrix and herb residue. The hydrophilicity of the reinforcement material decreased after it was treated with a silane coupling agent, and the compatibility between the treated reinforcement material and PB was improved [106].

The key operating parameters affecting the treatment, such as concentration of acid or alkali solutions, soaking time, and temperature, need to be optimised to have the most desired mechanical and physical properties. Increasing alkali concentrations have been linked to improved mechanical characteristics. However, exceeding the optimal concentration of chemical reagents may cause fibre degradation and have a detrimental impact on the tensile strength of composites. Gibeop et al. [79] revealed that alkali treatment with 3% NaOH concentration does not get rid of the amorphous material, with fibre pulling out holes in the PLA matrix, as shown by SEM images. On the other hand, jute fibres that have been treated with 5% NaOH concentration have good contact with the matrix, which makes the tensile strength better.

### 7.3. Biological Treatment

Biological treatment of fibres outcompetes chemical treatment without harsh chemicals or elevated temperatures. This treatment optimises the fibre surface for composite applications by using microorganisms, such as bacteria and fungi. These modifying agents are promising in developing composites with good mechanical properties that are both green and environmentally friendly. Enzymatic treatment is now gaining popularity, thanks to the high selectivity and specificity of enzymatic action that only targets the undesirable constituents without disrupting the structural modification of the important components [51,97].

Werchefani et al. [89] examined the impact of hemicellulases (cellulase-free xylanase) and pectinases on the alfa fibre surface, based on the hypothesis that hemicellulose and pectic components are accountable for moisture absorption and mechanical improvement. Their research demonstrated that these enzymes are excellent at improving the mechanical characteristics and water resistance of PLA composites. According to their findings, pectinase treatment was more effective than xylanase for eliminating undesirable materials, roughening the fibre surface, splitting alfa fibres into finer fibres, and enlarging surface accessibility for good polymer/filler interactions. As a result, an enhancement of tensile modulus and tensile strength was noticed when compared to that of unmodified samples. By getting rid of hemicellulosic and pectic components, enzyme treatments also make the surface less polar, which makes it less likely to absorb water.

The effects of three different enzymes (pectinase, laccase, and cellulase) on the reinforcing capability of bamboo fibres (BF) in poly(hydroxybutyrate-co-valerate) (PHBV) were studied by Zhuo et al. [102]. Melt blending of fibres and resin was followed by IM to fabricate the composites. All composites improved in mechanical properties following enzymatic treatment. However, the improvement was not significant. Pectinase had the best modifying impact of the three enzymes. The tensile strength, impact strength, flexural strength, and flexural modulus of PHBV composites with pectinase-treated BF increased by 4%, 7.1%, 6.2%, and 6.3%, respectively. They concluded that two factors contribute to the improvement of mechanical characteristics. The first is the surface roughness of BF, which is more favourable for stress transfer in composites. Second, the reduced polarity of BF after the removal of the OH group, lignin, and free cellulose on the surface. This feature is preferred for better compatibility with weakly polar PHBV and hence improves the interfacial compatibility of BF/PHBV composites. Werchefani et al. [89] reached the same conclusion: composites treated with pectinase had the best mechanical properties and the least amount of water absorption.

The combination treatment of xylanase and pectinase was conducted on DPF-reinforced PBS composites [45]. The highest tensile modulus (1600 MPa) was achieved with 20% of treated fibre reinforced composite, which was due to the synergistic effect of the two enzymes that impart the highest cellulose-rich fibre while degrading the amorphous components. The simultaneous action of both enzymes exposed more individual fibre bundles and cellulose microfibrils and reduced fibre diameters, which are believed to achieve the best mechanical properties. The efficacy of combined enzyme treatment was demonstrated by the depolymerisation of lignin, pectin, and hemicellulose during treatment with xylanase and pectinase, which verified an increase in the stiffness of the composites. Treated DPF has proven beneficial in a variety of applications where these mechanical properties are demanded. The study proved the combination of enzyme treatment benefits not only from the treatment efficiency but also from lowering the operational time. Another biological treatment using bacterial cellulase enzymes was applied to ramie fibres by Thakur and Kalia [103]. They used bacterial cellulases from two different bacterial strains (*Brevibacillus parabrevis* and *Streptomyces albaduncus*) to modify the surface properties of the PBS/ramie fibre biocomposites. The authors found that the ramie fibre surface is free from gums and polysaccharide layers and was cleaned and restructured to become more compatible with the hydrophobic PBS matrix. Therefore, there was better interlocking between the two phases, which helped the biocomposite to demonstrate satisfactory mechanical performance.

## 8. Advantages and Disadvantages of Reinforced Bioplastics and Its Treatment

Natural fibres derived from agricultural wastes serve as an ecological and cost-effective alternative to typical petroleum-based materials, since they substantially reduce the dependency on fossil fuels and greenhouse gas emissions. Depending on the plant source, the physical and mechanical characteristics of fibres can vary in terms of density, tensile strength, Young’s modulus and elongation at break (Table 2) [26,27,107,108]. As shown in Table 3, these fibres have significantly higher strength and stiffness values than bioplastics and conventional plastics [3,14,109,110,111]. Given the properties of natural fibre, it is preferred to reinforce polymers with high-strength natural fibres to produce natural fibre-reinforced bioplastics (NFRP). The addition of fibres derived from renewable and infinite resources reduced the overall cost of the composite material while improving the waste management techniques in a sustainable manner.

Potential avenues for improving reinforced bioplastic properties for a variety of applications are being explored. Fibre treatment is a novel approach for improving interfacial adhesion between fibre and matrix. Bioplastics that contain surface-treated fibre as a reinforcing agent increase fibre–matrix binding. The modification with NaOH alkaline solution, for example, splits the fibre bundles into finer fibres. The smaller fibres were impregnated with polymer material, enhancing the interface between the fibres and the matrix. The opening of fibre bundles and partial removal of cementing constituents results in rougher fibre surfaces, which facilitates matrix penetration into fibres. This suggests that the fibre-resin integration can have a significant impact on the stress transmission at the interface via mechanical interlocking [89]. Without the mechanical locking or formation of linkages within this region, the efficiency of the stress transfer mechanism is thought to be low, and the composite could not withstand the load when loaded. This phenomenon will be worsened if the reinforcement material does not disperse well throughout the composite, causing uneven load distribution [46].

Higher compatibility between fibres and matrix was expected with the inclusion of reinforcement, and the reinforced bioplastics demonstrated superior mechanical performance with modified fibre compared to composites made with untreated fibre [70,73,79,86]. Reinforcing surface-treated fibres allows the mechanical performance to be improved without deconstruction. The high specificity of enzymatic treatment allows them to target the non-cellulosic fibre components while retaining the natural structure of cellulosic fibres [42]. Added to that, treating fibre with the combined action of two enzymes contributes to a more fibrillose structure and enhanced stiffness of the reinforced bioplastic [45]. The enzyme-treated fibre-reinforced bioplastics have the lowest moisture absorption properties after eliminating the hydrophilic components on the fibre surface. The ability to resist moisture is beneficial in the preparation of composites for construction, automotive, and packaging applications [89].

Disadvantageously, some studies found a considerable decline in the elongation at break upon reinforcement of surface-treated fibres [45,59,99]. This is most likely owing to the reinforcing action of treated fibres, which limits the molecular movements, leading to stiffer and less flexible bioplastics [89,99]. However, as the fibre loading increased, the Young’s modulus decreased. This may be associated with the formation of aggregates at higher fibre content, leading to stress concentration zones and lower mechanical properties [78,86]. Another drawback of surface-treated bioplastics is the treatment parameters, which often deteriorate the mechanical performance of bioplastics when used in excess, as they damage the fibre surface [73,112,113]. As a result, optimal parameters and conditions should be carefully chosen to achieve the desired level of modification and boost the treatment efficiency.

## 9. Applications of Reinforced Bioplastics

Over decades, natural fibres have proved their excellence in substituting costly carbon and glass fibres. They have high specific tensile properties and lower density than synthetic fibres, making them lighter and more fuel efficient [43]. NFRP shows promise in a variety of areas, including automotives, aerospace, construction, consumer goods, protective equipment, packaging, and so forth. Because of their sensitivity to environmental degradation, NFRPs are currently limited to non-load-bearing interior components in civil engineering and automotive parts [91,114].

NFRPs have several advantages over conventional composites in the automotive industry, including increased acoustic insulation and mechanical properties, lower weight and manufacturing cost, recyclability, renewability, and eco-efficiency. They can be used to make door panels, seat cushions, armrests, and headrests [27,64]. Lighter composite parts used in vehicles to replace metal and heavier materials can lower transport weight, hence indirectly boosting fuel efficiency [115]. Werchefani et al. [89] fabricated biopolymer composites reinforced with alfa fibres from PLA. Mechanical testing shows that the composites have the required properties for interior automotive parts where composite strength is a necessity for performance.

By capitalising on its lower density, tool wear, and cost, natural fibre has surpassed synthetic fibre in many applications and is ideally suited for use as a reinforcement in polymer composites or cement matrices [58]. Indeed, natural fibre can be used to manufacture windows, doors, window frames, roof tiles, ceilings, and floor mats in the construction industry [27]. Sisal fibre and coir fibre have also been explored as roofing components instead of asbestos, which is carcinogenic [91,107]. Traditional composites have substantial pollution and disposal difficulties at the end of their useful life. As a result, there is a stronger desire to employ green products in order to leave a smaller environmental footprint.

When it comes to food applications, gas barrier properties (water vapour and oxygen permeability) are significant features to access the viability of materials because they affect the deterioration of moisture-sensitive products and their shelf-life [116]. In the PLA/PBS matrix, the presence of both cellulose nanocrystals (CNC) and surfactant-modified cellulose nanocrystals (s-CNC) provoked an improvement in oxygen and water vapour barrier properties [117]. In a prior study undertaken by Papadopoulou et al. [118], cocoa bean shells (CBS) as natural active additives in PLA composites were shown to represent a promising possibility for developing active food and biodegradable packaging materials by conferring antioxidant activity to the composites. Melt compounded PBAT/torrefied CG composites exhibited improved hydrophobicity, increased water contact angle values, and significant enhancement in the thermomechanical properties. Because of its high hydrophobicity, the biopolymer composite has the potential to be used in food packaging [119].

## 10. Conclusions

Bioplastics represent a new plastic generation, paving the way toward sustainability, renewability, and biodegradability. Their mechanical behaviour can be measured in terms of tensile, flexural, impact, and hardness. Reinforcing agents are added to bioplastics to strengthen their mechanical properties and expand their fields of application. For biocomposites, the choice of filler type, aspect ratio, filler loading, and surface treatment applied greatly influenced the final mechanical properties. To tailor the performance of final composites, uniform dispersion of reinforcement inside the matrix and a strong degree of interaction between them are required in composite materials. The hydrophilic fibre is modified for further compatibility enhancement with the hydrophobic behaviour of the polymer matrix. Treated fibres have a rough surface texture, which is critical for penetration into the matrix, enabling maximum stress transfer across the interface and a better mechanical interlocking system. Subject to the treatment strategies, most studies showed a better increment in fibre hydrophobicity, interfacial adhesion between fibre and matrix, and superior mechanical properties. The conditions and parameters used for surface treatments can cause changes in structure, morphology, and mechanical properties, consequently affecting the fibre-reinforced composites. Hence, proper fibre modifications enable better stimulation of their properties for usage as reinforcements in composites. A polymer composite with desired qualities that perfectly meets the requirements for a particular application can be fabricated by manipulating the fibre content, orientation, size, or manufacturing processes. Fibre-reinforced biocomposites find use in a variety of fields based on the qualities required. Further research on performance is needed to enlarge the domain of applications of biopolymer composites.

## Figures and Tables

**Figure 1 polymers-14-03737-f001:**
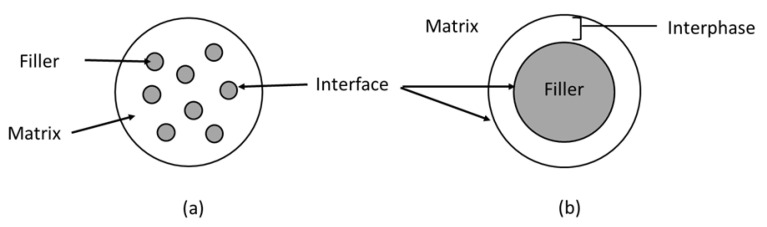
(**a**) Schematic of filler-matrix interface; (**b**) interphase region between the filler and matrix.

**Figure 2 polymers-14-03737-f002:**
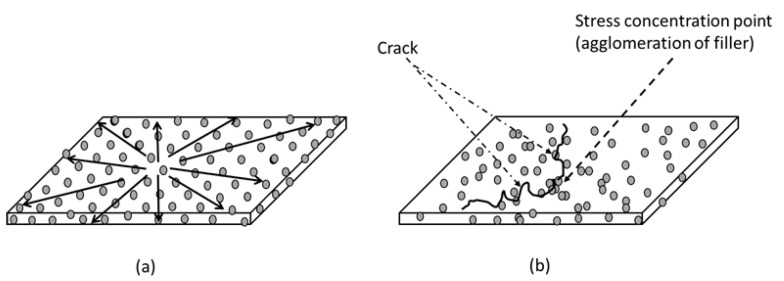
Load distribution for reinforced plastic (**a**) fillers are dispersed uniformly; (**b**) fillers are not well dispersed within the matrix.

**Figure 3 polymers-14-03737-f003:**
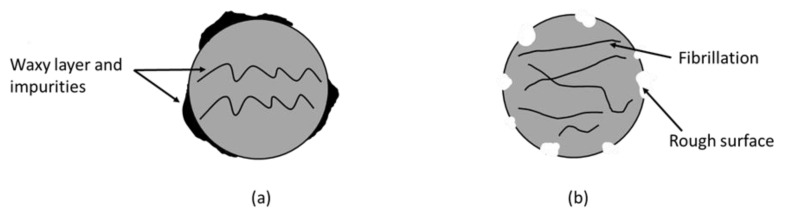
(**a**) The morphological change in the fibre surface before and (**b**) after surface treatment.

**Figure 4 polymers-14-03737-f004:**
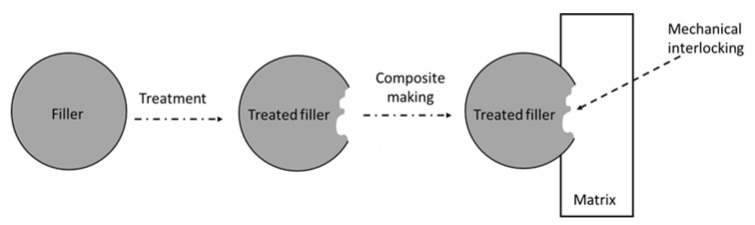
Formation of mechanical interlocking for a composite system.

**Figure 5 polymers-14-03737-f005:**
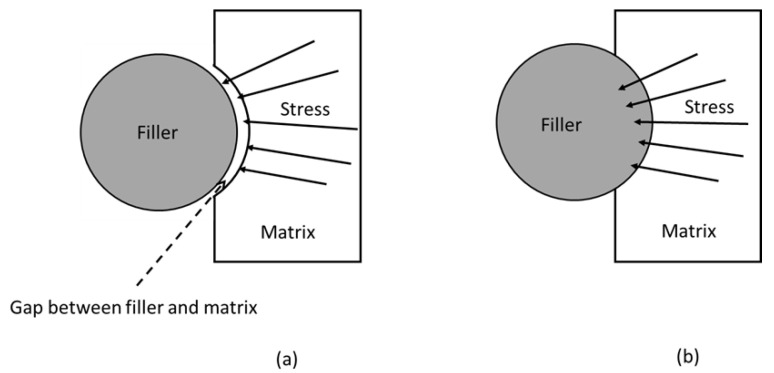
Stress transfer efficiency between filler (**a**) without interaction between filler and matrix (**b**) with interaction between filler and matrix.

**Table 1 polymers-14-03737-t001:** A summary of various surface treatments applied to fillers and their effects on the composite mechanical properties.

Treatment Type	Method	Filler	Matrix	Fabrication Method	Mechanical Performance	References
	Plasma treatment	Jute fibre	PLA	Injection moulding	Plasma polymerised jute fibre composites exhibited an increment in tensile strength, Young’s modulus, and flexural strength up to 28, 17, and 20%, respectively. The IFSS of plasma polymerised jute fibre composites exhibited a significant increase of 90% more than untreated fibre, with a value of 6.84 MPa.	[79]
Physical treatment	Plasma treatment	Coir fibre	TPS	Compression moulding	Oxygen plasma with stronger etching was more influential in all conditions compared to air plasma, with the composite’s tensile strength and elastic modulus increased by up to 300% and 2000%, respectively.	[73]
	Corona treatment	Miscanthus fibre	PLA	Extrusion- compression moulding	Lower fibre content (20% and 30%) increased Young’s modulus more than the greater fibre content (40%).	[78]
	Corona treatment	Date palm fibre	PLA	Extrusion- compression moulding	Significant improvement in tensile strength and Young’s modulus with 30% reinforcement of treated palm fibres in PLA, achieving the highest elastic modulus compared to untreated reinforcements and the PLA matrix.	[86]
	Electron beam irradiation	Bamboo powder	PLA	Injection moulding	The PLA/EBP5/ES 5phr composite demonstrated a 12% increase in notched impact strength over pure PLA.	[70]
Chemical treatment	NaOH alkali treatment	Alfa fibre	PLA	Injection moulding	The tensile strength and Young’s modulus of the composite were strengthened by 17% and 45%, respectively, when 20 wt% NaOH-treated alfa fibres were included.	[89]
	NaOH alkali treatment	Rice husk	TPS	Compression moulding	The composites developed from alkaline-treated RH at a 20 wt% concentration gave the highest tensile strength by a factor of 220%.	[88]
	Acetylation	Sugarcane fibre	TPS	Extrusion	The addition of AcSF to the composite mixture increased the product’s tensile strength while decreasing its water affinity.	[98]
	Acetylation and silanisation	Grape stalk powder	PBS	Injection moulding	Treated biocomposites showed better tensile properties than the control polymer. Acetylated GS powder gave the maximum improvement in Young’s modulus from 616 MPa to 732 MPa.	[59]
	Maleic anhydride, NaOH alkali, and salinisation	Palm fibre (Macaíba)	PCL	Injection moulding	PCL composites with 15% and 20% MA treated MF showed the highest elastic modulus among all the samples. MA treatment presented the best mechanical performance, whereas NaOH treatment resulted in the worst.	[99]
	Silanisation	Coffee husk	PBAT	Melt extrusion	The addition of 40 wt% silane-treated CH increased the composite’s mechanical properties (tensile strength, Young’s modulus, and elongation at break) as compared to the 40 wt% untreated CH-reinforced PBAT composite.	[72]
	Silanisation	Silicon carbide	PBAT/PC	Solution casting and melt extrusion	The PBAT/PC composite with T-SiC showed a substantial enhancement in tensile strength and Young’s modulus, with a reasonable drop in ductility.	[100]
	Maleic acid and silanisation	Coconut shell powder	PLA	Compression moulding	The treated composite’s tensile strength and Young’s modulus increased after the CS surface-treated with maleic acid and 3-APE coupling agent but had lower elongation at break.	[101]
	NaOH alkali treatment	Jute fibre	PLA	Injection moulding	Jute fibres treated with 5% NaOH concentration have good interaction with the PLA matrix, resulting in an improvement in tensile strength.	[79]
Biological treatment	Xylanase and pectinase enzymatic treatments	Alfa fibre	PLA	Injection moulding	The tensile strength of PLA/xylanase and PLA/pectinase composite samples is increased by ≈22% and ≈27%, respectively, when compared to that of unmodified samples.	[89]
	Xylanase and pectinase enzymatic treatments	Date palm fibre	PBS	Injection moulding	The combined action of two enzymes (xylanase and pectinase) gave the highest tensile modulus of reinforced composites (1600 MPa).	[45]
	Pectinase, laccase, and cellulase enzymatic treatments	Bamboo fibre	PHBV	Injection moulding	The values of tensile strength, impact strength, flexural strength, and flexural modulus were greatest for pectinase-treated bamboo fibre/PHBV composite.	[102]
	Cellulase enzymatic treatment	Ramie fibre	PBS	Compression moulding	The tensile and flexural strength of treated fibre reinforced biocomposites increased as the fibre concentration increased (0.5% to 1%).	[103]

**Table 2 polymers-14-03737-t002:** Properties of commonly used natural fibres to reinforce bioplastics.

Property/Natural Fibre	Bagasse	Bamboo	Cotton	Coir	Jute	Ramie	Oil Palm
Density (g/cm^3^)	1.25	0.6–1.1	1.5–1.6	1.2–1.5	1.3–1.5	1.5	0.7–1.55
Tensile strength (MPa)	222–290	140–800	287–800	140–180	200–773	400–938	248
Young’s modulus (GPa)	17	11–17	5.5–13	4–6	10–55	61.4–128	3.2
Elongation at break (%)	1.1	1.4	7–8	30	1.5–1.8	3.6–3.8	25

**Table 3 polymers-14-03737-t003:** Properties of bioplastics and conventional plastics.

Property/Polymer	PLA	PHA	PBS	PP	PET	PS
Density (g/cm^3^)	1.24	1.25	1.26	0.91	1.3–1.4	1.05
Tensile strength (MPa)	37–66	20–40	30–35	15–27	55–79	24–60
Flexural modulus (MPa)	2392–4930	1280–3668	-	850–1050	1000–2300	2100–3000
Young’s modulus (GPa)	2.7	2.95	0.27	0.95–1.77	2–4	3.4
Elongation at break (%)	0.5–9.2	1.4–5.5	8–13	100–600	15–165	1.6–2.5

## Data Availability

Not applicable.

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
