# Peer review of "Factors Affecting Mechanical Properties of Reinforced Bioplastics: A Review"

_polymers, 2022, doi:10.3390/polym14183737_

Round 1
Reviewer 1 Report (Previous Reviewer 2)
The manuscript deals with a review of the factors affecting the mechanical properties of reinforced bioplastics.
This review is interesting. Nevertheless, the manuscript has lack of depth and the authors must rearrange the manuscript and discuss in more detail the advantages and disadvantages of the different factors affecting the bioplastics and also point out new directions, rather than showing a compilation of several different studies. Moreover, the text should be balanced with the presented figures and tables.
The English language must also be revised.
Differences between reinforced bioplastics and common plastics should be highlighted.
Author Response
- Differences between reinforced bioplastics and common plastics should be highlighted.
Response: This is an excellent suggestion. We have revised the text to emphasize this point. Kindly refer to Section 8 (page 22, line 959-969), Table 2 and 3.
Reviewer 2 Report (New Reviewer)
With the environmental problems caused by non-degradable plastics, more and more attention has been paid to bioplastics. This review summarises the research on the properties of bioplastics modified by fibre reinforcement, with a focus on mechanical performance. This review gives a very comprehensive overview of the research progress of fiber reinforced bioplastics. I think this is a very good manuscript. But the manuscript need a minor revise before publish.
(1) The application of the bacterial cellulose in the bioplastics should be mention.
(2) Some references may be added in the manuscript which the DIO are10.3390/polym11030508, 10.3390/polym11121981, 10.3390/polym13101654, 10.3390/polym8040129.
Author Response
- The application of the bacterial cellulose in the bioplastics should be mentioned.
Response: Thank you for this suggestion. We have included the application of the bacterial cellulose in the bioplastics in the manuscript. Kindly refer to Section 4.3 (page 7, line 328-339).
- Some references may be added in the manuscript which the DIO are10.3390/polym11030508, 10.3390/polym11121981, 10.3390/polym13101654, 10.3390/polym8040129.
Response: Thank you for the suggestion. We have included some of the references in the manuscript.
Round 2
Reviewer 1 Report (Previous Reviewer 2)
The manuscript was improved.
This manuscript is a resubmission of an earlier submission. The following is a list of the peer review reports and author responses from that submission.
Round 1
Reviewer 1 Report
In my opinion, the manuscript entitled: "Factors Affecting Mechanical Properties of Reinforced Bioplastics: A Review - will not arouse much interest.
The title is promising, but the content is disappointing. The article presents a description of the factors influencing the properties of biopolymers. There is nothing interesting here, because the influence of factors on the mechanical properties is the same for biolopolymers and ordinary polymers. Perhaps the authors have noticed some spectacular cases of atypical action of factors influencing the properties? However, I found no evidence of this.
- The article would be more interesting if the authors took into account the biopolymers synthesized by microorganisms;
- Description of injection molding is based on articles about petroleum polymers;
- Abstract is written in an unacceptable form;
- And not very informative drawings.
The manuscript does not meet the journal's high standards. In my opinion it should be rejected.
Reviewer 2 Report
The manuscript deals with a review of the factors affecting the mechanical properties of reinforced bioplastics.
This review is interesting. Nevertheless, the authors must rearrange the manuscript and discuss in more detail the advantages and disadvantages of the different reinforced bioplastics, relation with fillers, and treatments and also point out new directions, rather than showing a compilation of several different studies. Moreover, the text should be balanced with the presented tables and figures.
The English language must also be revised.
Please separate values from units, e.g. “80 W” not “80W”.
Author Response
Please see the attachment, thank you

Round 2
Reviewer 2 Report
The manuscript was slightly improved. The authors failed to present and discuss in more detail the advantages and disadvantages of the different reinforced bioplastics, relation with fillers, and treatments and also point out new directions.